

# Stack of cellular lamellae forms a silvered cortex to conceal the opaque organ in a transparent gastropod in epipelagic habitat

Daisuke Sakai[1], Jun Nishikawa[2], Hiroshi Kakiuchida[3] and Euichi Hirose[4]

[1] School of Regional Innovation and Social Design Engineering, Kitami Institute of Technology, Kitami, Hokkaido, Japan

[2] Department of Marine Biology, School of Marine Science and Technology, Tokai University, Shimizu, Shizuoka, Japan

[3] Innovative Functional Materials Research Institute, National Institute of Advanced Industrial Science and Technology (AIST), Nagoya, Aichi, Japan

[4] Faculty of Science, University of the Ryukyus, Nishihara, Okinawa, Japan

Corresponding author
Euichi Hirose,
euichi@sci.u-ryukyu.ac.jp

## ABSTRACT

**Background**. Gelatinous zooplankton in epipelagic environments often have highly transparent bodies to avoid detection by their visual predators and prey; however, the digestive systems are often exceptionally opaque even in these organisms. In a holoplanktonic gastropod, *Pterotrachea coronata*, the visceral nucleus is an opaque organ located at the posterior end of its alimentary system, but this organ has a mirrored surface to conceal its internal opaque tissue.

**Results**. Our ultrastructural observation proved that the cortex of the visceral nucleus comprised a stack of thin cellular lamellae forming a Bragg reflector, and the thickness of lamellae (0.16 µm in average) and the spaces between the lamellae (0.1 µm in average) tended to become thinner toward inner lamellae. Based on the measured values, we built virtual models of the multilamellar layer comprising 50 lamellae and spaces, and the light reflection on the models was calculated using rigorous coupled wave analysis to evaluate their properties as reflectors. Our simulation supported the idea that the layer is a reflective tissue, and the thickness of the lamella/space must be chirped to reflect sunlight as white/silver light, mostly independent of the angle of incidence.

**Conclusions**. In *P. coronata*, the cortex of the visceral nucleus comprised multicellular lamellae that form a chirped Bragg reflector. It is distinct in structure from the intracellular Bragg structures of common iridophores. This novel Bragg reflector demonstrates the diversity and convergent evolution of reflective tissue using reflectin-like proteins in Mollusca.

# INTRODUCTION

In marine epipelagic habitats, many predators, such as fish, rely on their vision to forage for prey. To avoid detection by predators and prey, several strategies are utilized by pelagic

organisms to conceal their bodies (*Johnsen, 2014*). Some macroplankton, such as jellyfish and salps, often have gelatinous, transparent body as a defensive strategy. Furthermore, some of these plankton have nanostructures on their body surface, and these structures potentially reduce the reflection of light on their transparent body (*Hirose et al., 2015*; *Bagge, Osborn & Johnsen, 2016*; *Kakiuchida et al., 2017*; *Sakai et al., 2018*; *Sakai et al., 2019*; *Hirose et al., 2021*). However, as transparent bodies transmit not only visible light but also harmful ultraviolet radiation, there is a trade-off between threats from predation and ultraviolet radiation (*Hansson, 2000*; see also *Johnsen & Widder, 2001*). Moreover, it is difficult to make some organs transparent, even in transparent zooplankton. For instance, the stomach lining and the contents are usually opaque (brownish or sometimes bluish) and often conspicuous in a bright background in salps. These opaque organs are usually smaller compared to their body size, and minimizing the size of the opaque mass in transparent animals may reduce the risk of detection by predators/prey with a visual system. Mirrored surfaces are another strategy for concealing opaque organs or bodies; as the light field underwater approximately homogenous to the vertical axis, the intensity of light reflected by a vertical mirror is not much different from the intensity behind the mirror (*e.g.*, *Denton, 1970*; *Johnsen & Sosik, 2003*). Accordingly, light reflection on the mirrored surface conceals opaque tissue below the mirror.

Pterotracheoidea consists of holoplanktonic gastropods that float or swim slowly moving the fin in the open sea. In this superfamily, where the members of Atlantidae usually have a transparent shell that can hold their entire body, the species of Carinariidae have a very small shell that covers only a small part of the body, and the species of adult Pterotracheidae have no shells at all. They are often quite large and may be easily found by their natural enemies. Considering their soft, gelatinous bodies, transparency is an adaptation to avoid predators, particularly in Carinariidae and Pterotracheidae. Transparency is also important for foraging for prey, as they are carnivorous. Their bodies are so transparent that predated animals in the alimentary tract can be identified from outside the body (*Seapy, 1980*). The visceral nucleus is exceptionally opaque in the bodies of carinariids and pterotracheids. It is a cone-shaped organ located at the end of the alimentary tract; reddish-brown digestive gland cells surround the tract, and the anus opens at the apex of the cone. Although this organ is held in a small shell in carinariids, pterotracheids lack this shell. In an aquarium, *Pterotrachea* spp. usually swim with their ventral side facing upward, and the visceral nucleus remains vertical (*Seapy & Young, 1986*). As a result, the apex of the cone-shaped organ, that is, the anal opening, remained downward. This posture minimizes the visible area of the visceral nucleus when a predator or prey looks at the animal from above or below. Moreover, as the visceral nucleus organ often appears metallic silver in live specimens, the reflected light may conceal the organ in the epipelagic layer. Electron-dense platelets are present in the silver cortex of this organ (*Seapy & Young, 1986*). These platelets are assumed to have a high refractive index. Alternating layers of high and low refractive indices result in a highly reflective structure, which is a Bragg reflector (*Land, 1972*). Reflection using Bragg reflectors has been reported in various organisms, including mollusks such as cephalopods (*Kawaguti & Ohgishi, 1962*; *Mirow, 1972*; *Cooper, Hanlon & Budelmann, 1990*) and bivalves (*Kawaguti, 1966*; *Land, 1966*).

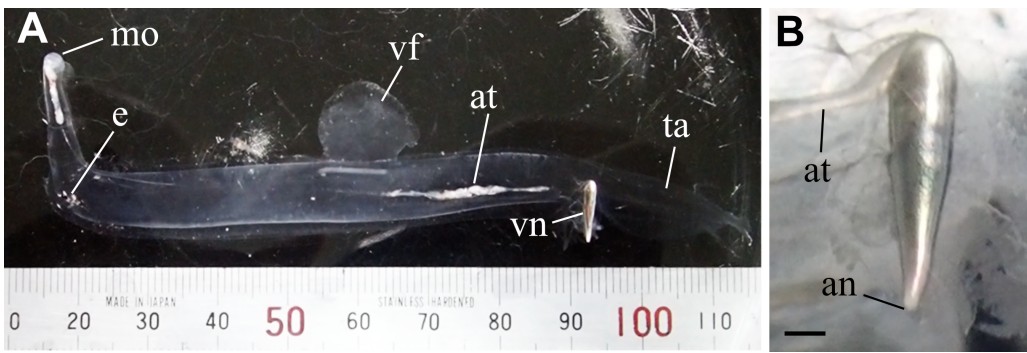

**Figure 1** **Live specimen of *Pterotrachea coronata*.** (A) A whole body with a scale (minimum scale, 1 mm). Upside is ventral. (B) An enlargement of the visceral nucleus. an, anus; at, alimentary tract; e, eye; mo, mouth; ta, tail; vf, ventral fin; vn, visceral nucleus. Scale bar = 1 mm. Photographed by J. Nishikawa.

*Pterotrachea coronata* Forskål, 1775 is a pterotracheid gastropod distributed mainly in the epipelagic layers of tropical and subtropical waters (*Taylor & Berner, 1970*; *Aravindakshan, 1973*; *Mutaf, Akşit & Gökoğlu, 2008*; *Clark et al., 2021*). Its silver visceral nucleus is conspicuous throughout its transparent body (Fig. 1). In the present study, fine structures of the cortex of the nucleus were examined to describe the morphological basis of this highly reflective tissue. Based on the fine structures, we simulated the light reflection of the cortex using rigorous coupled wave analysis (RCWA) to evaluate the properties of light reflection on this exceptionally opaque organ in a transparent pelagic gastropod.

## MATERIALS & METHODS

### Animals
*Pterotrachea coronata* were collected with the ORI net (mouth diameter 1.6 m, mesh size, 335 µm or 4 mm) on May 11, 2019 (34°52′57.0″N, 138°38′34.0″E) and on May 29, 2021, in Suruga Bay, Japan (34°42′16.5″N, 138°35′57.5″E). They were fixed in 2.5% glutaraldehyde−0.45 M sucrose−0.1 M cacodylate (pH 7.4) onboard and stored at 4 °C. The refractive index of the fixing solution was 1.360–1.361, as measured with a pocket refractometer PAL-RI (Atago Co., Ltd., Bellevue, WA, USA). The total body lengths of the fixed animals were 13.5, 8.3, 3.8, and 3.2 cm, respectively. Individuals for reflectance spectroscopic analysis were collected using the same net on January 19, 2022 (35°01′55.5″N, 138°33′05.2″E). Individuals were photographed at the laboratory of Tokai University and transported alive to the laboratory of AIST (Nagoya) for the measurement of reflection spectra.

### Reflectance spectroscopy
Reflectance spectra were measured at the parts of the live specimen including and excluding the visceral nucleus, and the colors of reflected light were estimated for comparison between the parts. The reflectance was spectrally measured using an optical setup assembled with a halogen light source (HL-2000; Ocean Optics, Inc., Largo, FL, USA), optical fibers (P100-2-UV–VIS; Ocean Optics, Inc.), integrating sphere detector (ISP-50-8-R-GT; Ocean

Optics, Inc., Largo, FL, USA), and spectrometer (Flame-S; Ocean Optics, Inc., Largo, FL, USA). A light beam with a diameter of 10 mm was normally incident on the specimen located five mm from the detector aperture. The normal specular reflectance through an optical aperture within a divergence angle of 11° was measured at wavelengths between 400 and 800 nm with a resolution of 0.4 nm. The calibration of the integrating sphere detector was carried out using a hemispheric reflectance standard (SRS-99-010; Labsphere, Inc., North Sutton, NH, USA) for 100% reflection and an ultralow reflection sheet (Acktar Metal Velvet #12-695; Edmund Optics Inc., Barrington, NJ, USA) for 0% reflection. This reflectance standard has a Lambertian reflection surface with more than 99% hemispheric reflectance at every visible wavelength, and it was closely stuck to the optical aperture of the integrating sphere for the detector calibration. The measured values do not represent the absolute values of reflectance, but the reflected light spectrum captured by the integrating sphere is considered to maintain its original spectrum. The chromaticity was calculated from the reflectance spectrum based on the CIE (Commission Internationale de l'Eclairage) standard solar illuminant D65 and expressed in CIE 1931 using the chromaticity diagram template installed in Origin 2021b (OriginLab Corp., Northampton, MA, USA).

## Electron microscopy

The visceral nuclei were excised from the specimens in the fixatives with razor blades. Following a brief rinse with 0.45 M sucrose–0.1 M cacodylate (pH 7.4), the specimens were post-fixed in 1% osmium tetroxide buffered with 0.1 M cacodylate (pH 7.4) for 1.5 h at 4 °C. After dehydration through an ethanol series, the specimens were cleared with $n$-butyl glycidyl ether, and embedded in an epoxy resin. Sections of 0.5–1 µm in thickness were stained with 1% toluidine blue for light microscopy. Thin sections were double-stained with uranyl acetate and lead citrate; they were examined using a transmission electron microscope (TEM; JEOL JEM-1011) at 80 kV. We used ImageJ 2.1.0 to measure the length of some electron micrographs.

## Simulation of light reflectance on the cortex of the visceral nucleus

To evaluate the optical properties of the Bragg structure in the cortex of the visceral nucleus, we built the virtual models of the Bragg structure based on the ultrastructural observations and calculated the light reflection on the models with RCWA using DiffractMOD software (RSoft Design Group, Inc., Ossining, NY, USA). In this simulation, the refractive indices were defined as 1.34 for the layers with a lower refractive index, assuming that the index is similar to that of seawater. The index of the higher refractive index layer was assigned as 1.44, considering the indices of the Bragg reflectors found in cephalopods (*Brocco & Cloney, 1980*; *Ghoshal et al., 2014*). The refractive indices from 1.34 to 1.54 were also tested to clarify how the difference in the refractive index affects the reflectivity. Both transverse electric wave (TE wave) and transverse magnetic wave (TM wave) were used as the light source, ranging from 300 to 800 nm in wavelength, while the wavelength was fixed as 480 nm and 589 nm (D-line) in some simulations. The major lights are assumed to be around 480 nm at the depths where these animals are typically found (*e.g.*, *Herring, 2002*).

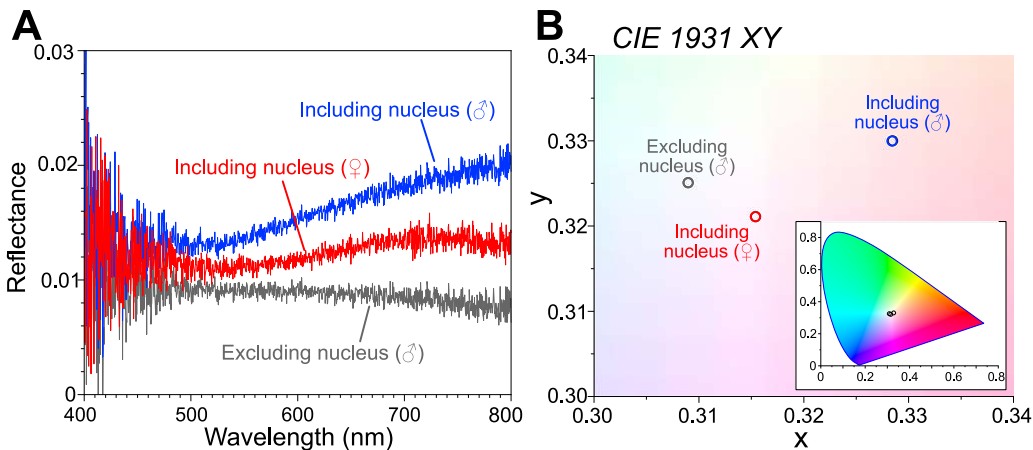

**Figure 2  Reflectance spectra and chromaticity of the visceral nucleus.** (A) Reflectance spectra on the circular areas, 10 mm in diameter, including the visceral nucleus (blue line, male; red line, female) and excluding the nucleus (gray line, male). The values are averages of multiple measurements. (B) Chromaticity calculated from the reflectance on the CIE 1931 diagram. Inset, chromaticity in the full range of the diagram.

Based on the calculated spectral reflectance of the virtual models and the CIE standard solar illuminant D65, we calculated the chromaticity of the reflected light at incident angles of 0°, 20°, 40°, 60°, and 80° as aforementioned in reflectance spectroscopy.

## RESULTS

### Gross morphology and reflectance spectra of the visceral nucleus

The visceral nucleus is an elongated, cone-shaped organ located at the end of the alimentary tract. In live specimens, the nucleus reflects the incident light well and appears metallic silver, whereas the entire body, including the alimentary tract, is transparent except for the alimentary contents (Fig. 1A). Upon closer observation of the nucleus, the cortical tissue is reflective and the medullar tissue appears reddish brown (Fig. 1B).

The reflectance spectra are shown as an average of the following measurements (Fig. 2A): the male body including the visceral nucleus ($n = 3$; blue line), the male body excluding the visceral nucleus ($n = 5$; gray line), and the female body with the visceral nucleus ($n = 8$; red line). The measured reflectance exhibits the difference depending on the parts of the bodies, although it is very small ($<0.03$) even in the part including the nucleus. Such low reflectance arose from our experimental technique, that is, the measured area (approximately 10 mm diameter) is largely occupied by the transparent tissue of the animal placed on the ultralow reflection sheet. Moreover, the specimen surfaces were curved, and some of the reflected light can be undetected. The chromaticity of the reflected light was calculated from the measured spectra and plotted in the CIE 1931 (Fig. 2B). The reflected light from the area including the nucleus was faintly reddish for the female specimen (smaller) and slightly more brownish for the male specimen (larger). In the area excluding the nucleus, the intensity of the reflection was small, and the color was almost neutral, *i.e.*, white.

## Histological observation of the visceral nucleus

The metallic reflectivity of the visceral nucleus had reduced in the fixed specimen; however, the nucleus still had a glistening surface (Fig. 3A). The alimentary tract entered this cone-shaped organ on the lateral side of the basal part, and the anus opened at the apical end of the cone. In histological sections stained with toluidine blue, the visceral nucleus consisted of three parts: the alimentary tract, medulla, and cortex (Figs. 3B, 3C). The medulla was a loose cell mass that surrounded the epithelial wall of the alimentary tract. The thickness of the medulla differed depending on the part of the nucleus; the medulla near the anal opening was usually thinner than the medulla at the middle part. The cortex formed a thick, multilamellar wall packing around the tract and medulla. The thickness of the cortex also varied depending on the parts of the nucleus and size of the individual; it was approximately 10–20 µm thick in the present specimens.

The major part of the cortex was a multilamellar layer; lamellae of high electron density were arranged parallelly (Fig. 3D). There was a squamous epithelium over the multilamellar layer, and some free cells were occasionally found between the epithelium and outermost lamella. Beneath the multilamellar layer, several fusiform cells loosely formed a stack of cells that were collectively assigned as a sub-lamellar layer in this report. In the section located near the anal opening in Fig. 3D, the medulla was poorly developed, and the sphincter muscle surrounded the ciliated epithelium of the alimentary tract.

## Fine structures of the cortex of the visceral nucleus

In the multilamellar layer, 30–50 electron-dense lamellae were parallel to the surface of the organ (Fig. 4A). There was always a void space between the lamellae, and the electron-dense lamellae did not split or attach to the neighboring lamellae. The number of lamellae depended on the total thickness of the multilamellar layer. Thin cells with nuclei were occasionally found in the lamellae. The peripheral part of the cell was so thin that it was almost indistinguishable from electron-dense lamellae (Fig. 4B). The lamellae varied in thickness and appeared thicker in the outer part of the layer.

The observation at a higher magnification suggested that each lamella was a very thin, flattened cell with electron-dense cytoplasm; the lamella was always membrane-bound and contained fragments of rough endoplasmic reticulum (rER) (arrowheads in Fig. 4C) and a mitochondrion-like structure with two membrane layers (asterisks in Fig. 4C). The multi-lamellar layer entirely covered the medulla and alimentary tract; however, the layer was sometimes interrupted, and irregularly shaped cells filled the gap (Fig. 4D). At the terminus, the edge of each lamella always ended as a roundish tip and never folded back (Fig. 4E), indicating that a single cell had formed each lamella.

The sub-lamellar layer is a loose stack of fusiform cells that did not attach to each other (Fig. 5A). Since these sub-lamellar cells appeared fusiform in both the sagittal and cross sections of the visceral nucleus, they possibly had a ravioli-like shape with stretched cell peripheries. The border between the multi-lamellar and sub-lamellar layers was indistinct because the thickness of the fusiform cell peripheries was almost the same as the thickness of the electron-dense lamellae. The cytoplasm of these cells was filled with rER (Fig. 5B),

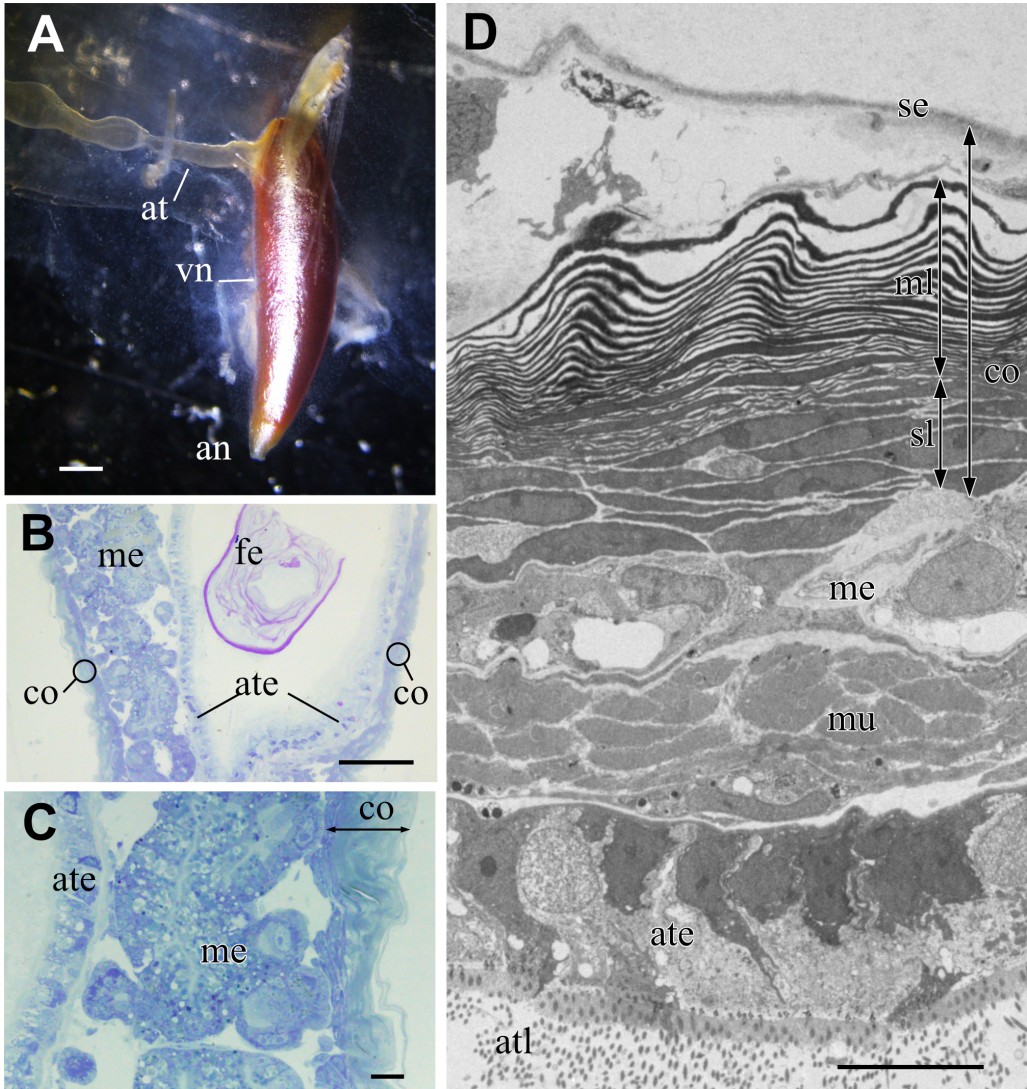

**Figure 3 Visceral nucleus of *Pterotrachea coronata*.** (A) External view of the visceral nucleus (vn) with the glistening surface (fixed specimen). (B) Histological section of the visceral nucleus near the anal opening. (C) Enlargement of B. (D) TEM image of the neighbor section. an, anus; at, alimentary tract; ate, epithelium of the alimentary tract; atl, lumen of the alimentary tract; co, cortex; fe, feces; me, medulla; ml, multi-lamellar layer; mu, sphincter muscle; se, squamous epithelium; sl, sub-lamellar layer. Scale bars: 1 mm in A, 100 μm in B, 10 μm in C and D.

suggesting high synthetic activity of proteins in the cells, and rER was usually found even in the thinly stretched cell periphery (Fig. 5C).

## Simulation of light reflectance with RCWA

The thickness of each lamella and space of the multilamellar layer in Fig. 4A was measured and plotted from the innermost to the outermost lamella/space (Fig. 6, Table S1). Based on these measurements, we built three simplified models of the multilamellar layer comprising 50 lamellae and spaces to calculate the reflectance. In the even model (blue lines in Fig. 6),

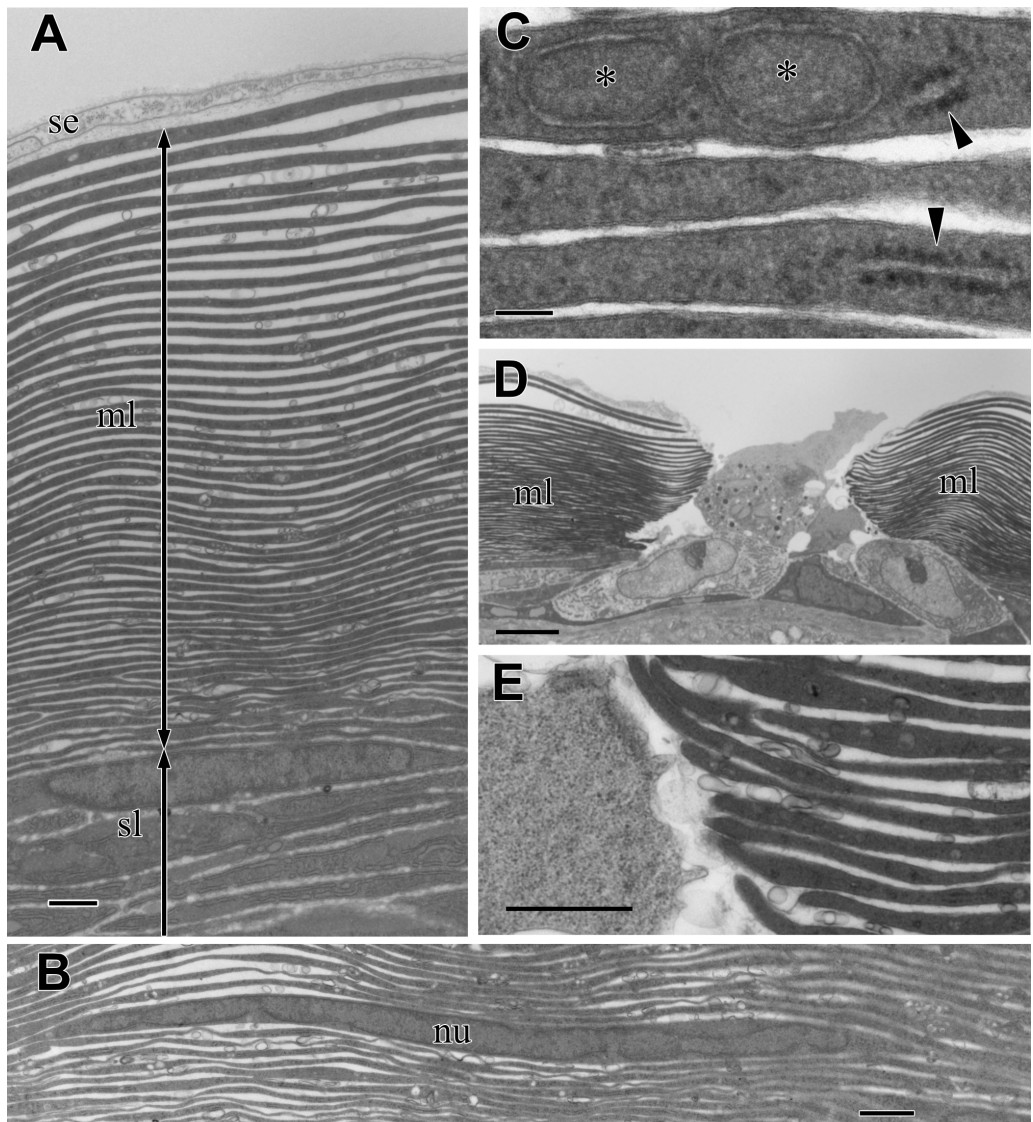

**Figure 4  TEM images of the multi-lamellar layer (ml) in the cortex at the middle part of the visceral nucleus.** (A) Cortex. (B) Electron-dense lamellae and a nucleated cell in the multi-lamellar layer. (C) Enlargement of the electron-dense lamellae containing mitochondrion-like structure (asterisks) and rough endoplasmic reticulum (arrowheads). (D) Terminus of the multi-lamellar layer. (E) Enlargement of C. nu, nucleus; sl, sub-lamellar layer; se, squamous epithelium. Scale bars: 1 μm in A, B, and E; 0.1 μm in C, 5 μm in D.

all lamellae were 0.16 μm thick, and all spaces were 0.1 μm wide. In the up-chirp models (red lines in Fig. 6), the thickness of the outermost lamella was 0.27 μm, thinning by 0.004 μm, and the innermost lamella was 0.05 μm in thickness. Similarly, the outermost space was 0.15 μm wide, thinning by 0.002 μm, and the innermost space was 0.05 μm in width. In the down-chirp model (not shown), the thickness of the outermost lamella was 0.05 μm thick, increasing by 0.004 μm up to 0.27 μm in thickness. The outermost space was 0.05 μm wide, increasing by 0.002 μm up to 0.15 μm width. For all models, the

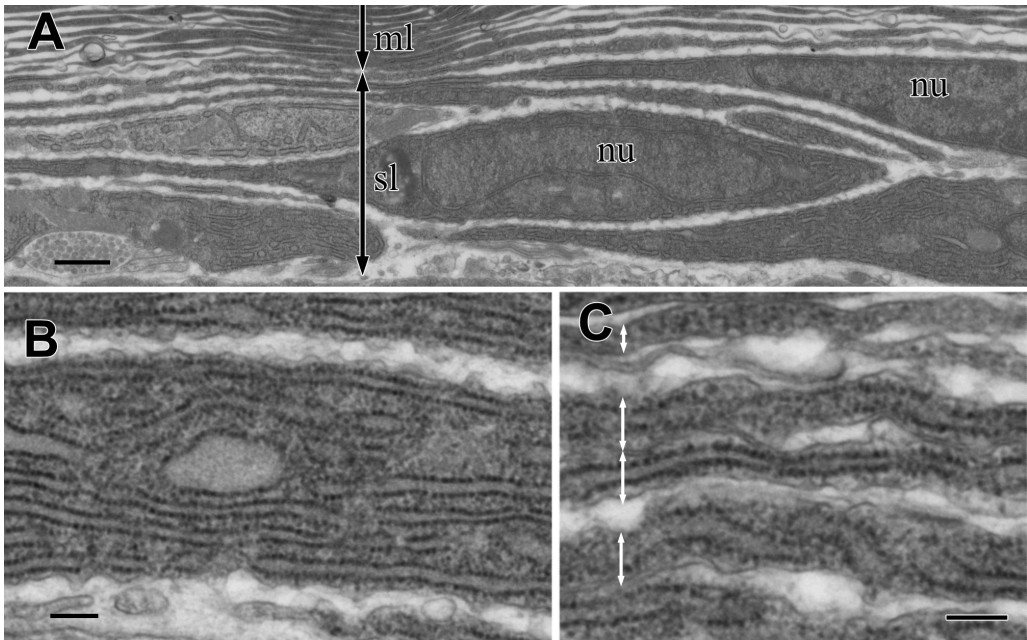

**Figure 5** **TEM images of the sub-lamellar cells in the cortex.** (A) A border between the multi-lamellar layer (ml) and the sub-lamellar layer (sl). (B) Cytoplasm of the sub-lamellar cell filled with rough endoplasmic reticulum. (C) Elongated part of the sub-lamellar cells. Two-way arrows indicate each cell. nu, nucleus. Scale bars: 1 μm in A; 0.2 μm in B and C.

thickness of the multilamellar structure was 13 μm, and the up-chirp model was the closest to the multilamellar layer of the visceral nucleus among the three models. In addition, we built 'thick models' to test the potential effects of shrinkage due to fixation of the live specimens. In the 'thick models', the thickness of lamellae and spaces was set to 125% of the original models described above, assuming that the live specimens had shrunk to 80% by the fixation and the following processes for electron microscopy.

The reflectance determined by the incident angle and refractive index difference was similar for light at 480 nm (Fig. 7A) and 589 nm (Fig. 7B). The difference in refractive index between the lamella and space ($\Delta n$) in the three models of the multilamellar structure, and the reflectance was greater when $\Delta n$ was larger. The range of incident angles where the reflectance is greater than 0.5 is much narrower for the even model (Fig. 7, left column) than for the up-chirp (Fig. 7, middle column) and down-chirp model (Fig. 7, right column), although the peak reflectance of the chirp models was smaller than that of the even model. In the three models, the reflectance of the TE wave was always greater than that of the TM wave, for which the reflectance was zero at Brewster's angles ranging from 45° ($\Delta n = 0$) to approximately 49° ($\Delta n = 0.2$). The reflectance in the up-chirp model was almost the same as that in the down-chirp model, and the maximum difference between these two models was approximately 0.00051 (TE wave) and 0.0002 (TM wave) for 480 nm light and 0.00024 (TE wave) and 0.00013 (TM wave) for 589 nm light.

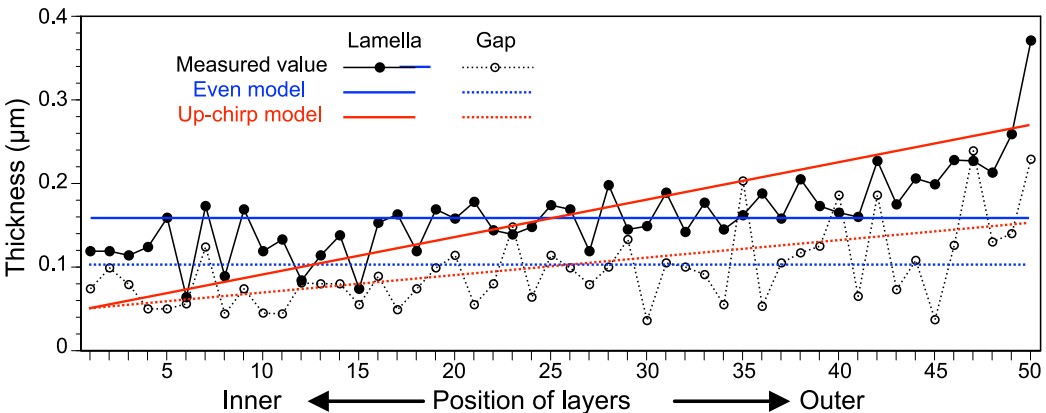

**Figure 6 The thickness of lamellae (closed circles, solid lines) and spaces (open circles, broken lines) measured from the electron micrograph of the multi-lamellar layer.** Blue and red lines indicate the thickness of lamellae (solid lines) and spaces (broken lines) in the virtual models for the simulation of light reflection.

When $\Delta n$ was assumed to be 0.1, a strong reflectance over 0.9 was found in a particularly narrow range of wavelengths depending on the incident angle in the even model (Fig. 8, left column). In contrast, light of a wide range of wavelengths was reflected in the up-chirp (Fig. 8, middle column) and down-chirp models (Fig. 8, right column), while the peak reflectance of the chirp models was smaller than that of the even model. The reflectance in the up-chirp model was almost the same as that in the down-chirp model, and the maximum difference between these two models was approximately 0.00030 for the TE wave and 0.00085 for the TM wave in the simulation. In all three models, the reflectance of the TE wave was always greater than that of the TM wave, for which the reflectance was zero at approximately 47° (Brewster's angle). It is estimated that the color of the reflected light of the CIE standard solar illuminant D65 changes significantly in the even model as the angle of incidence changes (Fig. 9A, Fig. S1A). However, the color was almost white at various incident angles in the chirp models (Fig. 9B, Fig. S1B). The simulation results for the 'thick models' were similar to those in the original models, while there was a slight shift in the reflection distribution (Fig. S2).

The reflectance properties were compared among the up-chirp models with different numbers of lamellae: the one-lamella layer was the outermost lamella of the up-chirp model, the five-lamella layer was a stack of lamellae from the outermost to the fifth lamella, and so on. The reflectance was greater when the number of lamellae increased for both TE and TM waves (Fig. 10). In all models, the reflectance of the TE wave was greater than that of the TM wave, for which the reflectance was zero at approximately 47° (Brewster's angle). The effect of the lamellar number on the reflectance varied depending on the incident angle and wavelength, and some cases are shown (Fig. 11, Table S2): 440 nm (40° and 60°), 480 nm (40°, 60°, 80°), 500 nm (50°), 600 nm (60°). Among these TE waves, 440 nm light transmits deeper into the water column than any other wavelength of visible light in the

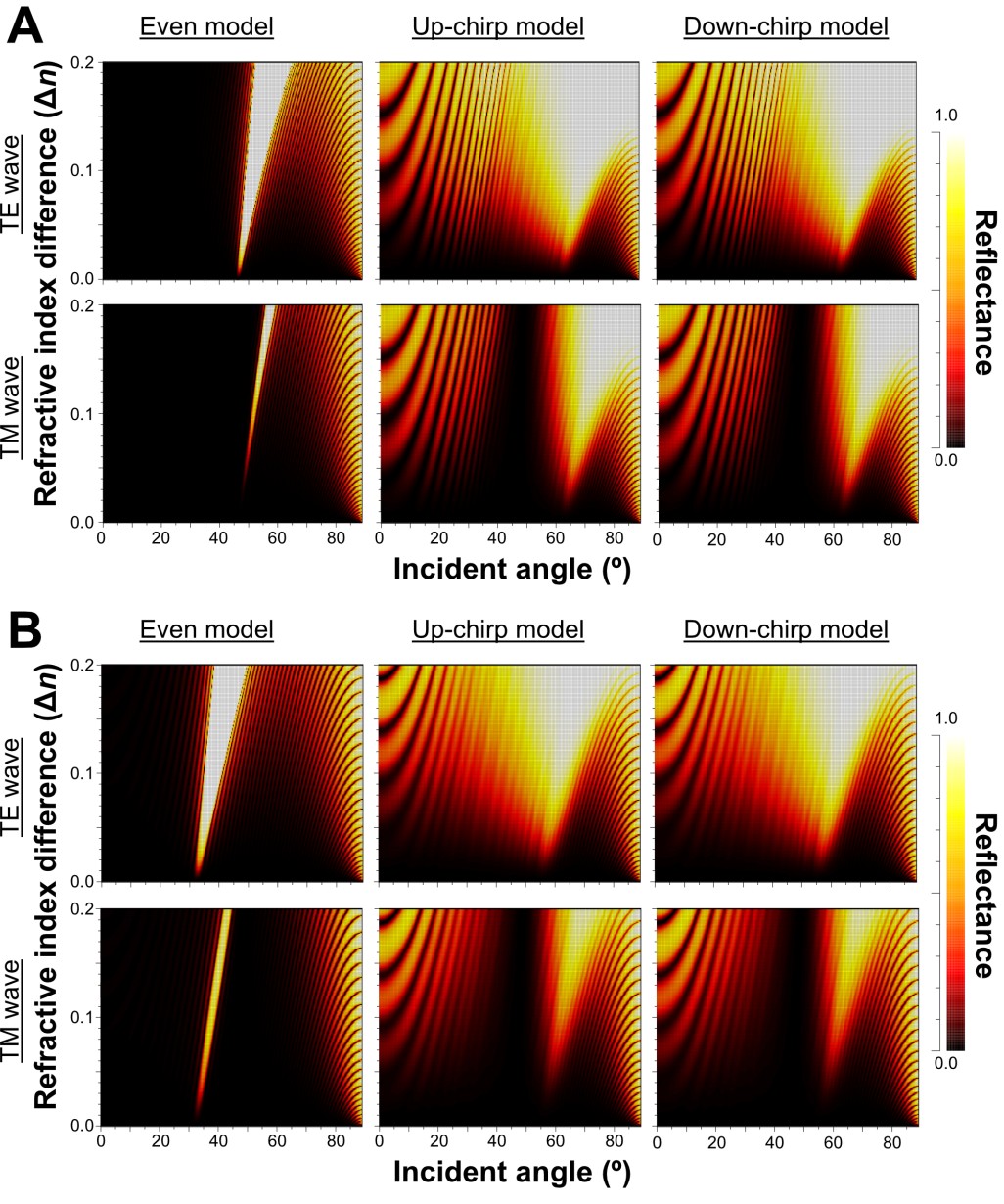

**Figure 7** Simulation results of the reflectance determined of 480 nm light (A) and 589 nm light (B) by the incident angle and the difference in refractive index between the lamella and space ($\Delta n$) in the even model (left), up-chirp model (middle), and down-chirp model (right). No reflection occurs at Brewster's angle for TM (45°, $\Delta n = 0$; ca. 49°, $\Delta n = 0.2$).

open sea, and the major lights are assumed to be around 480 nm at the depths where these animals are typically found (*e.g.*, *Herring, 2002*).

As the outer lamellae tended to be thicker than the inner lamellae in microscopic observation, we compared the reflectance of the five-lamella layer, a stack of lamellae from the outermost to the fifth lamella, between the up-chirp and down-chirp models. The

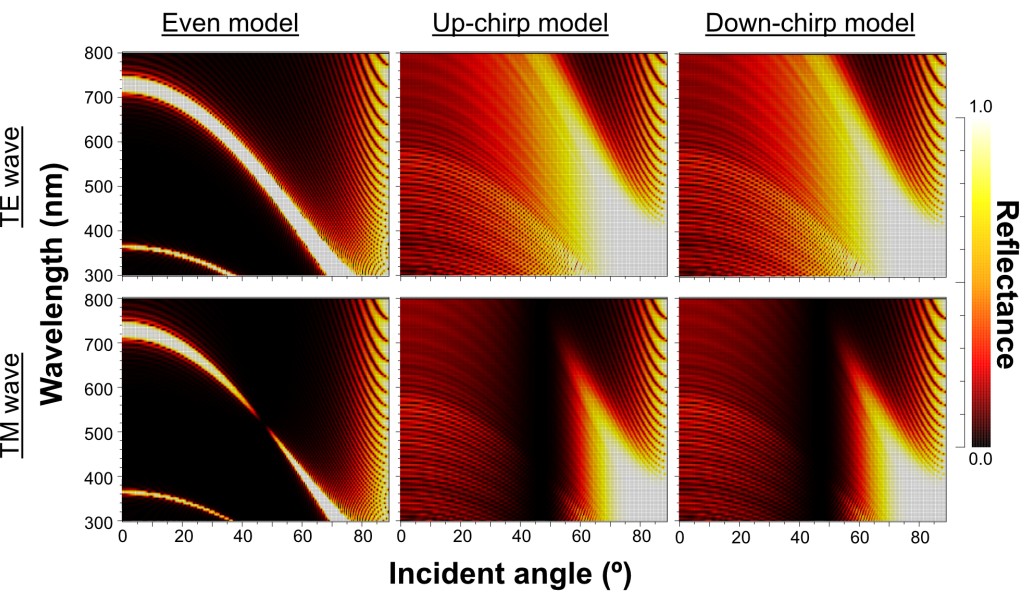

**Figure 8** **Simulation results of the reflectance determined by the incident angle and the wavelength in the even model (left), up-chirp model (middle), and down-chirp model (right) for TE wave (upper row) and TM wave (lower row).** The difference in refractive index between the lamella and space ($\Delta n$) is 0.1 in these simulations. No reflection occurs at approximately 47° (Brewster's angle) for TM.

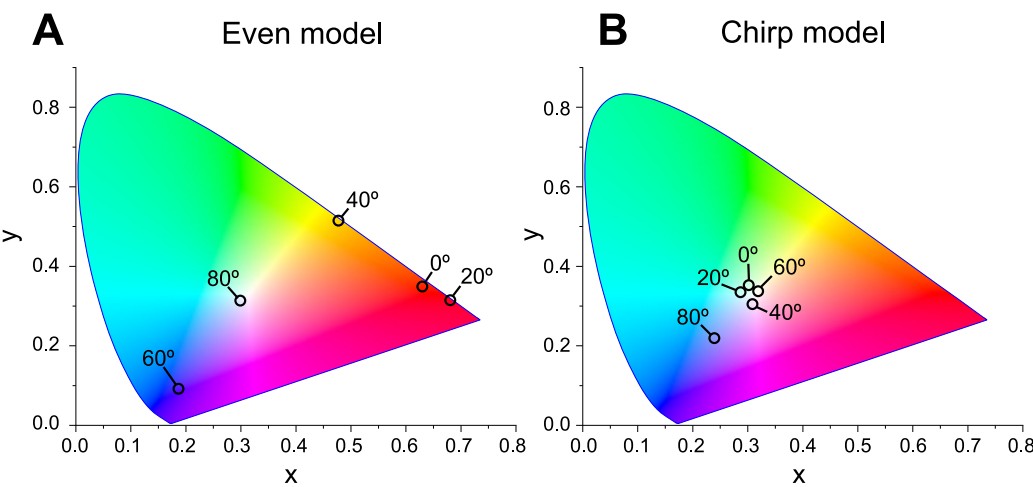

**Figure 9** **Chromaticity of the reflected light at incident angles of 0°, 20°, 40°, 60°, and 80° on the CIE 1931 diagram in even (A) and chirp mode (B).** The light source of the calculation is CIE standard solar illuminant D65.

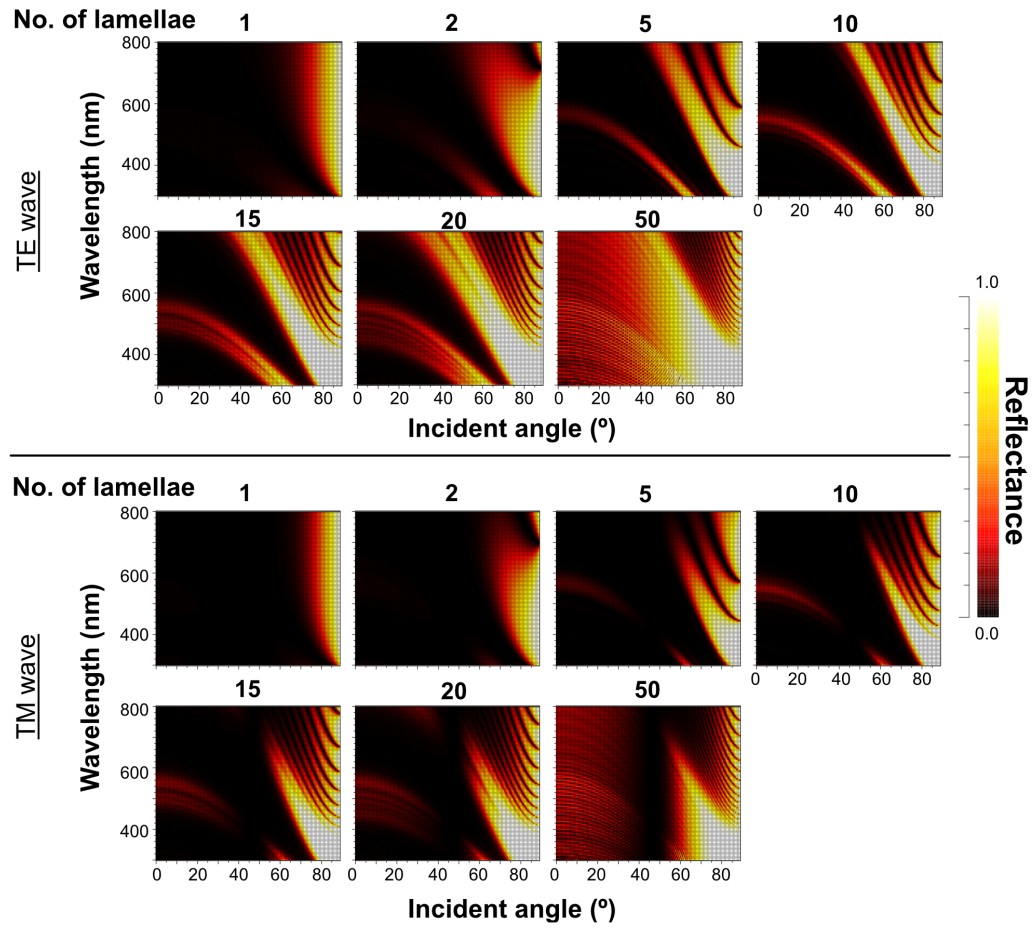

**Figure 10** Simulation results of the reflectance of the up-chirp model with different numbers of lamellae for TE wave (**upper rows**) and TM wave (**lower rows**). No reflection occurs at approximately 47° (Brewster's angle) for TM.

five-lamella layer of the up-chirp model reflected light more than that of the down-chirp model for both TE and TM waves (Fig. 12).

## DISCUSSION

### Silvered visceral nucleus

The visceral nucleus of *P. coronata* appears metallic silver in live specimens because the cortex of the nucleus strongly reflects the incident light, and the mirrored surface is expected to conceal opaque organs or bodies behind it (*e.g.*, *Denton, 1970*; *Johnsen & Sosik, 2003*). Upon closer observation, the nucleus has a slight reddish-brown tinge, owing to the color of the medullar tissue of the nucleus, which is consistent with the results of the reflectance spectra (Fig. 2). The reflected light of the visceral nucleus is broadband and similar in color and intensity to ambient light, making less visible to the predators or prey of *P. coronata*.

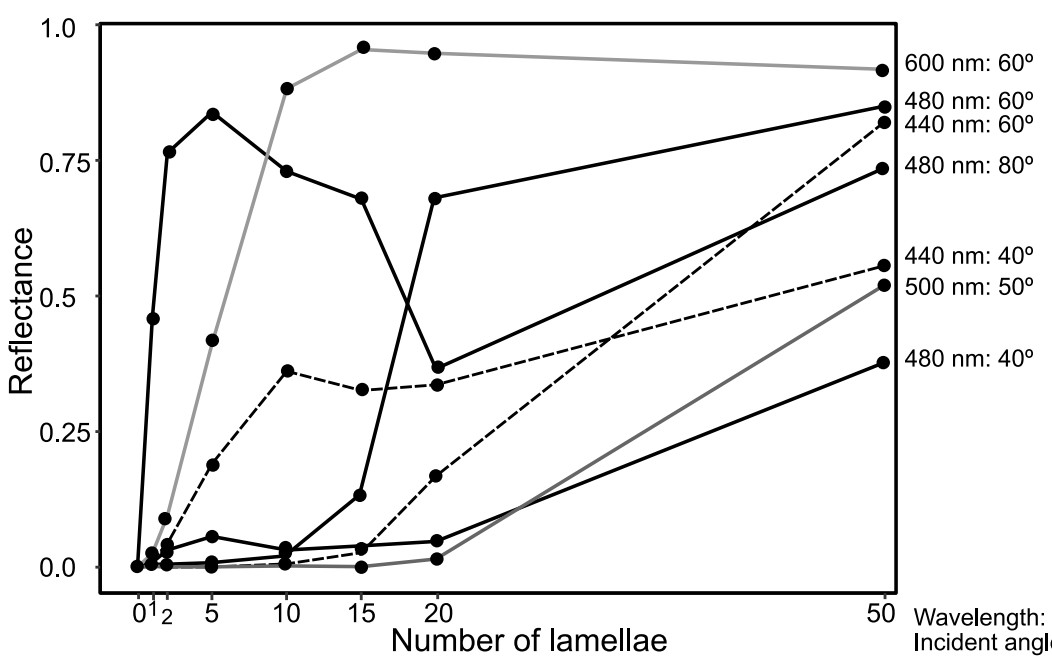

**Figure 11** Plots of reflectance of TE wave in the up-chirp models with different numbers of layers at particular wavelength and incident angles.

## Multi-lamellar layer of the cortex in the visceral nucleus

In the visceral nucleus of *P. coronata*, the reddish-brown medulla was entirely covered by the silvered cortex. The cortex mainly consists of a multilamellar layer and a sub-lamellar layer, and the former is assumed to be responsible for the high reflectivity. The multi-lamellar layer in *Pterotrachea hippocampus* Philippi, 1836 was described as a stack of "long flat platelets" or "iridophore platelets" in the reflective layer of its visceral nucleus (*Seapy & Young, 1986*), and these platelets were possibly recognized as intracellular structures in iridophores as known in the iridophores of other mollusk species (*Kawaguti & Ohgishi, 1962*; *Kawaguti, 1966*; *Land, 1966*; *Mirow, 1972*; *Cooper, Hanlon & Budelmann, 1990*). In the present study, each lamella was membrane-bound and contained mitochondrion-like structures and fragments of rER (Fig. 4C), suggesting that each electron-dense lamella originated from a cell, and the space between the lamellae was an intercellular space. Although mitochondrion-like structures are found in the cytoplasm, no cristae remain in them; it is doubtful whether these structures are functional. In the present study, the lamellae were neither split, fused with neighboring lamellae, nor folded back at the terminus, indicating that each lamella is a single cell. Therefore, the multilamellar layer of the visceral nucleus can be regarded as a stack of cellular lamellae. According to the descriptions of the platelets in *P. hippocampus* (*Seapy & Young, 1986*), a stack consists of approximately 35 platelets, and the thickness of the platelets ranges from 122 to 328 nm in the external portion of the layer, 91 to 244 nm in the middle portion, and 36 to 183 nm in the inner portion. These values are not very different from the values measured in *P.*

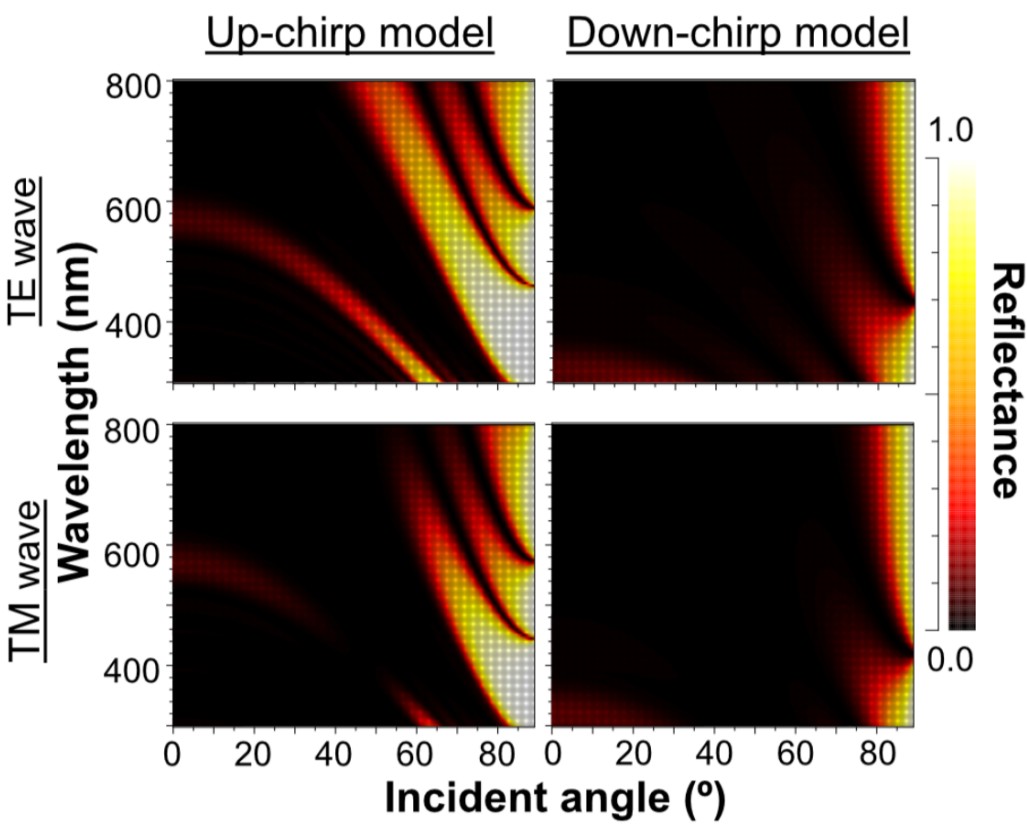

**Figure 12** The reflectance of the five-lamella layer (from the outermost lamella to the fifth) of up-chirp model (left) and down-chirp model (right) for TE wave (upper row) and TM wave (lower row).

*coronata* (Fig. 6), and the stacks of lamellae/platelets are up-chirped toward the inside in these two species.

The sub-lamellar layer is a loose stack of rER-rich cells with a thinly stretched cell periphery. Since the thickness and electron density of the cell peripheries are similar to those of electron-dense lamellae, the cell peripheries and lamellae are often indistinguishable. The sub-lamellar cells likely differentiate into lamellae in the multi-lamellar layer, and thus, the lamellae are supposedly added from the sub-lamellar layer, following the growth of the organ. The process of lamellae formation may explain why the thickness and number of lamellae vary among different parts within the organ and among individuals. The rER-rich cytoplasm of the sub-lamellar cells indicates active synthesis of proteinous substance(s) that is likely related to reflectance, such as a substance with a high refractive index. Since neither secretory activity, nor granular/vesicular formation is ultrastructurally found in these cells, the substances are assumed to accumulate in the cytoplasm. This may be the reason for the high electron density in the cytoplasm of the cellular lamellae and sub-lamellar cells.

## Cellular lamellae as a Bragg reflector

The most representative reflector in animals is a stack of alternating layers of high and low refractive indices (*Land, 1972*), *i.e.*, the Bragg reflector. Fish iridophores contain a stack of

platelets consisting of purine crystals, such as guanine and hypoxanthine. The refractive index of guanine is approximately 1.83 (*Land, 1972*). Bragg reflectors using purine platelets have also been reported in scallops (*Land, 1966*) and copepod (*Chae & Nishida, 1994*; *Chae et al., 1996*). Iridophores contain a stack of electron-dense platelets as Bragg reflectors in cephalopods (*Kawaguti & Ohgishi, 1962*; *Mirow, 1972*; *Cooper, Hanlon & Budelmann, 1990*) and giant clams (*Kawaguti, 1966*), and platelets have been identified with reflectins, a protein with a high refractive index, in the squid *Euprymna scolopes* Berry, 1913 (*Crookes, Ding & Huang, 2004*) and *Doryteuthis opalescens* (Berry, 1911) (*DeMartini et al., 2015*). The refractive index of the platelets varies from 1.42 in the octopus, *Octopus dofleini* (Wülker, 1910) (*Brocco & Cloney, 1980*) to 1.56 in the squid *Loligo forbesii* Steenstrup, 1856 (*Denton & Land, 1971*), and 1.44 was proposed as the refractive index of the reflectin-based platelets based on direct measurement in the squid *Doryteuthis opalescens* (Berry, 1911) (*Ghoshal et al., 2014*).

The stack of platelets is generally an intracellular structure in the molluscan iridophores, and the layers of high and low refractive indices are the reflectin platelets and the cytosol, respectively. In contrast, in the squid *D. opalescens*, the plasma membrane of iridophores deeply invaginates to form a Bragg stack in which the layers of high and low refractive indices are cellular processes containing reflecting platelets and intercellular spaces, respectively (*DeMartini, Krogstad & Morse, 2013*). In any case, the refractive index of the cytoplasm and intercellular space is assumed to be about the same as that of seawater, *i.e.*, about 1.34. The multilamellar layer of *P. coronata* is structurally similar to the latter Bragg stack; each cellular lamella forms the layer of high refractive index and intercellular gaps correspond to the layer of low refractive index. The reflectance of the fixed specimens is lower than that of the raw samples (Figs. 1A & 3B), probably because the refractive index of the fixing solution (1.36) is higher than that of seawater (1.34), resulting in a smaller difference in refractive index between the lamellae and gaps. The most significant difference between the iridophore and the multilamellar layer is that the iridophore in *D. opalescens* is iridescent as a single cell while multiple cells are required to form iridescent layer in *P. coronata*. In the silver eye tissue of *D. opalescens*, a homogeneously packed spindle-shaped cells forms a Bragg stack (*Holt et al., 2011*), and this is comparable to the multilamellar layer of *P. coronata* as a multicellular Bragg reflector. These multicellular stacks may also be structurally and functionally similar to the skin of the hatchetfish *Argyropelecus* sp. where an array of elongated cells orients light reflection (*Rosenthal, Holt & Sweeney, 2017*).

The metallic body colors of insects are formed by a stack of alternating layers with high and low refractive indices in their chitinous cuticle (*Neville, 1977*; *Parker, Mckenzie & Large, 1998*). Similar structures have also been found in reflectors of some crustaceans (*Parker, 1999*; *Nishida, Ohtsuka & Parker, 2002*). As the thickness of the cellular lamellae and spaces are up-chirped toward inside of the organ in *P. coronata*, the chitinous Bragg reflector in these insects and crustaceans are also up-chirped.

## Properties of light reflection on the virtual models of the stack of cellular lamellae

We calculated the light reflection of the three virtual models (even model, up-chirp model, and down-chirp model) based on the ultrastructures of the stack of cellular lamellae in *P. coronata*. Among the three, the up-chirp model was closest to the multilamellar layer of the visceral nucleus. The sequence of lamellae in the down-chirp models are oppositely arranged than in the up-chirp model, which is unnatural for the present animal but useful for considering why the thickness and pitch of the Bragg reflectors tend to decrease from the outermost layer (reflection surface) toward the inside, *i.e.*, up-chirp, not only in this mollusk but also in some arthropods (*Neville, 1977*; *Parker, Mckenzie & Large, 1998*; *Parker, 1999*; *Nishida, Ohtsuka & Parker, 2002*). In the simulation of all the models, the reflectance was greater when the difference in refractive index between the lamella and space ($\Delta n$) became larger (Fig. 5). Although the refractive index of the lamellae in the multi-lamellar structure is unknown in *P. coronata*, we assumed the refractive index of the lamellae as 1.44 in the simulations here. The basis of this assumption is the refractive index of the reflector, made of reflectin, in iridophores of some cephalopods, which is estimated to be 1.42–1.56 (*Denton & Land, 1971*; *Brocco & Cloney, 1980*; *Ghoshal et al., 2014*). We also consider the refractive index of the spaces to be similar to seawater or cytosol (1.34); thus, the difference in refractive index between the lamella and space ($\Delta n$) is assigned as 0.1 in all three models.

The properties of reflectance are significantly different between the even model and the two chirp models, whereas the properties are almost the same in the up-chirp and down-chirp models (Figs. 7 and 8). The light of a particular wavelength is strongly reflected in the even model, and the difference in reflected wavelength depends on the incident angle in the even model. This indicates that the color of the reflected light on the even model continuously changes, similar to an oil film on the water surface, when the angle of illumination and/or the observers' position is shifted (Fig. 9A). The spectrum of ambient light is limited by depth due to light absorption by seawater. In this condition, the even models can be easily detectable, because they will reflect light at limited angles of incidence. In contrast, light with a wide range of wavelengths is reflected in the chirp models, indicating that the reflected light is almost white (Fig. 9B). This is consistent with the color of the reflected light from the visceral nucleus, *i.e.*, silver (Fig. 1B) and may support the validity of our model-based simulations. The chirped multi-lamellar layer is considered as an adaptive character for *P. coronata* to conceal the opaque organ in water column. The virtual models used here were based on the ultrastructures of the multilamellar layer that possibly shrunk through the fixation and preparation process for electron microscopy. Since the layer spacing is thought to have a significant effect on reflector properties in the Bragg stack, we also performed the simulation for 'thick models,' which assume that the specimens observed in TEM had shrunk to 80% from the original live specimens. The results of both simulations were similar (Fig. 8 and Fig. S2), although a slight shift in the reflection distribution was observed. Thus, even if there was tissue shrinkage in some extent, it does not negate the discussions above.

The reflectance of the multilamellar layer increased with the number of lamellae (Fig. 10). It is difficult to determine the minimum number of lamellae for camouflage because

the change in reflectance differs depending on the wavelength of light and its incident angle (Fig. 11). Since the number of lamellae in the multi-lamellar layer is 30–50 in *P. coronata* (present observation) and about 35 in *P. hippocampus* (*Seapy & Young, 1986*), these numbers are probably enough to form a silvered cortex.

Our numerical simulations using RCWA showed agreement in optical reflectance between the up- and down-chirped periodic structures (Figs. 7 and 8), which is consistent with the optical reciprocity in light propagation (*e.g.*, *Hecht, 1987*). However, Bragg reflectors described in animals are often up-chirped toward inside. As for *P. coronata*, microscopic observations suggested that electron-dense lamellae were added to the inner end of the multilamellar layer, indicating that the thicker outer lamellae were formed first. Therefore, *P. coronata* forms an up-chirped lamellae sequence from the outermost lamellae to the inner lamellae. In Fig. 12, the reflectance of the five-lamella layer of the up-chirp model is greater than that of the down-chirp model, suggesting that forming a thicker lamella is adaptive for efficient light reflection even when the number of lamellae is small, such as in young individuals. Moreover, because thicker lamellae are expected to be more robust than thinner lamellae, it is beneficial for *P. coronata* to have thicker lamellae first, considering their function as a cortical shell of the visceral nucleus. Accordingly, the present results support the idea that the up-chirped sequence of lamellae in the multi-lamellae layer is functionally adaptive for *P. coronata*.

## CONCLUSIONS

In *P. coronata*, the silvered cortex of the visceral nucleus comprised a stack of cellular lamellae with a high electron density, that is, a multilamellar layer. The cellular lamellae are expected to have a higher refractive index than the intercellular space to function as Bragg reflectors in *P. coronata*. The cellular lamellae probably concentrate a proteinous substance with a high refractive index, which may be reflectin or related proteins found in the iridophores of cephalopods. This new Bragg reflector in *P. coronata* may demonstrate the diversity and convergent evolution of reflective tissue using reflectin-like proteins in Mollusca. The simulations of light reflection on the virtual models using RCWA support that the up-chirped sequence of the electron-dense lamellae is important for reflecting white light to conceal the opaque organ in the epipelagic water column. The combined study of comparative structural surveys and wave analyses of reflective tissues in various organisms potentially promotes the discovery of novel functional structures and mechanisms from nature.

## ACKNOWLEDGEMENTS

The authors thank the captains and crews of RV Hokuto and RTV Bosei (Tokai University) and scientists onboard for their support during the cruise. This study was performed as part of the Suruga Bay Research for Understanding Marine Ecosystems (SURUME) Project. We thank all members of the project for helping with the sampling.

### Funding

The present study was supported by KAKENHI (No. 20K06213) from the Japan Society for the Promotion of Science and a grant-in-aid from the School of Marine Science and Technology at Tokai University. There was no additional external funding received for this study. The funders had no role in study design, data collection and analysis, decision to publish, or preparation of the manuscript.

### Grant Disclosures

The following grant information was disclosed by the authors:
Japan Society for the Promotion of Science: 20K06213.
School of Marine Science and Technology at Tokai University.

### Competing Interests

The authors declare there are no competing interests.

### Author Contributions

- Daisuke Sakai performed the experiments, analyzed the data, prepared figures and/or tables, and approved the final draft.
- Jun Nishikawa conceived and designed the experiments, authored or reviewed drafts of the article, sample collection and preparation, and approved the final draft.
- Hiroshi Kakiuchida performed the experiments, analyzed the data, prepared figures and/or tables, and approved the final draft.
- Euichi Hirose conceived and designed the experiments, performed the experiments, analyzed the data, prepared figures and/or tables, authored or reviewed drafts of the article, and approved the final draft.

### Data Availability

The raw measurements for Fig. 6 and the reflectance data calculated using RCWA for Fig. 11 are available in the Supplemental Tables.

### Supplemental Information

Supplemental information for this article can be found online at http://dx.doi.org/10.7717/peerj.14284#supplemental-information.

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
