# Peer review of "Stack of cellular lamellae forms a silvered cortex to conceal the opaque organ in a transparent gastropod in epipelagic habitat"

_PeerJ, doi:10.7717/peerj.14284_

## Round 0.1 · original submission · Major Revisions

In my opinion, this manuscript needs revision in order to update the interpretation of the cellular architecture, reconsideration of the optical modeling to be more concordant with the very limited experimental measurements, and include wider relevant literature.

Reviewer 1 ·

Basic reporting

I'm very excited to read this paper, I have always been curious what the structure underlying heteropod reflectivity is!

The images are really nice, and it seems important to publish this novel structure from a novel taxon - I don't off the top of my head know of other examples of published gastropod iridescent structures, and it is really exciting for the field of optical structure in animals to see structures from a new taxon.

However, the analysis has some shortcomings.

Experimental design

There is a somewhat glaring mismatch between the limited reflectance measurements and semi-involved reflectance modeling. The measurements show a more or less broad-band reflectance but with a maximum intensity of ~3% and no real angular information. It is a small, squishy, fragile tissue that is hard to measure, and this (low) intensity is almost certainly an artifact. The corresponding modeling shows high reflectances of 100% and a strongly angle-dependent character. Given the experimental data available (TEM images and some likely faulty, low-resolution reflectance data), it seems sufficient to say "the reflectance is observed to be broad band, and the structure is consistent with broad-band reflectance". The further analysis of up-chirp and down-chirp stacks seems a little superfluous given the limited data on a hard system, and ultimately hurts the credibility of the paper.

Validity of the findings

The authors seem to think that the optics in squid iridocytes are entirely cellular, though I don't entirely understand this paragraph (around lines 321-331). However, the structure of squid iridocytes is very similar to what they propose here - the low-refractive index portion of the optics is extracellular fluid, while the high-refractive index portion is intracellular protein platelets. In some examples, (e.g., dorsal iridocytes in Loligo/Doryteuthis, see DeMartini et al. 2013) the intracellular platelets are wrapped around the extracellular fluid such that interference can arise from the platelets of a single cell. In other examples (e.g., the silver eye reflector of the same animal), interference arises from the extracellular fluid separating single cells (Holt et al., 2011).

So in fact, the cellular architecture of the heteropod reflectors seems identical to those of squid, a very interesting finding! It is difficult to tell from the included images whether optical interference arises from layers within a single cell (like the Doryteuthis dorsal reflectors) or from layers formed by several cells (like the Doryteuthis silver eye reflectors) but either is a valid comparison to known squid structures within the same animal.

In fact, the inner "flattened ravioli" structures observed here in the heteropod silver structure are strikingly similar to the "layered spindles" present in the squid silver eye reflector. The paper describing that structure (Holt et al. 2011) is not cited here, so the authors may not be aware of it.

Similarly, the outer layers of the heteropod structure look very similar to that described in a hatchetfish (Rosenthal et al. 2017). That paper speculates that the outer layers of the hatchetfish reflective structures may contribute to beam-shaping of a reflected search beam, it is interesting to speculate about a similar optical role for the heteropod.

Additional comments

I support publishing the novel structures documented here with updated interpretation of the cellular architecture and a reconsideration of the optical modeling to be more concordant with the very limited experimental measurements of a very tricky tissue.

·

Basic reporting

Line 45: “Johnsen” is spelled wrong in the citation

Lines 48-49: citations for antireflection seem heavily focused on the authors’ own work and exclude a quite relevant one that predates all but one of these (Bagge, L. E., Osborn, K. J., and S. Johnsen (2016) Nanostructures and monolayers of spheres reduce surface reflections in transparent hyperiid amphipods. Current Biology 26, 3071-3076).

Line 51: citation here misses one of the few citations on how transparent gelatinous zooplankton address UV damage (Johnsen, S., and E. A. Widder (2001). Ultraviolet absorption in transparent zooplankton and its implications for depth distribution and visual predation. Marine Biology 138, 717-730)

Line 53: is the stomach lining itself brown or is the coloration entirely due to the contents?

Line 57: merely concealing an opaque surface with a mirrored one will not camouflage it. The authors should explain here how mirror camouflage works in the ocean. See pioneering papers by Eric Denton on this topic.

Line 63: it might also be worth mentioning that the pterotracheids are often quite large and thus easier to see

Lines 74-75: again, it would be good to explain how this camouflage works

Line 80: I would say “bivalve” and not “clams”, especially since some of the best mirrors in bivalves are found in scallops

Line 132: this citation may also be useful for cephalopod Bragg stacks, though I’m unsure if it directly measures index. I might also check papers on iridophores by Lydia Mathger or Roger Hanlon (Holt, A., Sweeney, A. M., Johnsen, S., and D. E. Morse (2011). A highly-distributed Bragg stack with unique geometry provides effective camouflage for Loliginid squid eyes. Proceedings of the Royal Society: Interface 8, 1386-1399.)

Line 218: I’m pretty sure the delta symbol is not supposed to be italicized

Lines 209, 211 and 214: the numbers here have an astonishing number of significant figures, which doesn’t seem appropriate

Line 220-221: your measurements use percent reflectance and your model uses fractional reflectance. I’d use the same for both

Line 227: you can’t measure reflectance in degrees. I assume this is a typo

Lines 203-260: this section goes on for quite a while and in a form that can be hard to follow. I’d suggest a table or something else that makes the various parameter sweeps easier to follow.

Experimental design

Line 109: was this a specular reflectance standard? One can’t use a diffuse white standard for this. If it was specular, I’d make it clear to avoid confusion.

Lines 111-113: I’m uncertain why chromaticity was calculated, especially with a D65 illuminant. The relevant viewers will not be human and the illuminant will not be D65. If the authors want to say the reflectance is spectrally neutral, a reflectance spectrum would be more useful.

Lines 116-123: one issue with biophotonics is that the fixation and dehydration steps for TEM may shrink the tissues and thus change the dimensions of the Bragg stacks and alter the model. Have the authors accounted for this?


Line 136: why is 589 nm used? At the depths these animals are typically found, there is very little 589 nm light.

Line 139: again, I am unsure why human-based chromaticity with a terrestrial illumination spectrum is being used for an animal that lives in a nearly monochromatic blue environment and is not viewed by humans.


Line 160: as I mentioned earlier, fixation can change the dimensions of the Bragg stacks, which present a problem for modeling. Given that the reflectivity in fixed specimens is said to be reduced here, this confirms that this is an issue.

Validity of the findings

Line 148 and Figure 2: these reflectance values are so low as to be uninterpretable. 1-2% is darker than most objects that we would call black. I agree with the authors that some of the light would have reflected off the curved surface in other directions, but then why use a 10 mm beam? I’ve seen a large number of living specimens of these animals and never have seen one where the gut was wider than 10 mm (though it can be longer than 10 mm). However, it’s not clear how big a region was being read from. Was it also 10 mm in diameter? If so, then the reflectance measurements are mostly of other body parts. If it’s smaller, then it should be given in the methods, but to accurately measure the reflectance of a shiny curved surface, the spot that is being recorded from must be extremely small.

Line 180: can you be sure of the spacing of the lamellae given fixation artifacts?

Line 266: it’s unclear how the mirrored reflectance makes something hide against bright backgrounds. You need to discuss the theory of how mirror camouflage works in cylindrically symmetric light fields – as per Eric Denton’s work in the 1960s.

Lines 270-271: there is essentially no red light at the depths where these animals are found, so it’s impossible to see the color at all.

Line 357: one thing about the even model is that the underlying assumption is perfect evenness, which leads to highly variable reflectance depending on angle and wavelength. But a real ‘even’ reflector of the design seen here would have a lot of natural variance in spacing, since it is not a hard substance like the chitin in a butterfly scale. So it’s not quite fair to compare it to the chirped model. I would give the ‘even’ model some amount of variance in spacing of layers to give a better comparison with chirped.

Line 371: but we know the wavelength of most light in the sea is around 480 nm and we can guess at some good viewing angles for a predator approaching one of these animals

---

## Round 0.2 · Minor Revisions

Thank you for your careful attention to the reviewers comments. Reviewer 1 is satisfied with your responses and requests no further adjustments. Please see the comments on your revision from reviewer 2. All three comments should be addressed in your minor revisions.

Comment 1 requires the addition of a statement regarding the unreliable/suspect nature of the absolute reflectance numbers.

Comment 2 requires either a replacement of the relevant analysis with one at 480 nm or the addition of an analysis at 480 nm to make the discussion relevant to the light that these animals experience.

Comment 3 requires either the suggested sensitivity analysis for 20% shrinkage or a stronger argument for why this is not necessary should be provided as the current argument is insufficient. Shrinkage cannot be discounted in this situation or organism.

·

Basic reporting

I am happy with the revisions

Experimental design

I am happy with the revisions

Validity of the findings

I am mostly happy with the revisions but have three remaining issues:

1. since the authors are using a diffuse reflectance standard to measure specular reflectance, they can get odd results, for example a reflectance that appears to be greater than 100%. Since the authors have stated that they are not concerned about the absolute value of the reflectance, this is not a critical issue, but they should make it very clear in the text that that a number of factors render their absolute reflectance numbers suspect, so that readers will not use using them for their own papers.

2. I understand that 589 nm (the sodium D-line) is a natural and common wavelength for physicists and engineers. However, it really has no meaning for a relatively deep aquatic animal (and for the biologists that study them), where nearly all the light is at 480 nm. The same issue occurs with using human-based chromaticity, and I do not understand why confirming the human-observed color of the mirrored-gut is necessary. I suppose it depends on the intended audience. If this will mostly be read by physicists, then I wouldn't see a need for change, but it seems like this paper might mostly be read by marine biologists. How hard would it be to also analyze things at 480 nm?

3. I think the authors are downplaying the effects that fixation may have on the spacing of the Bragg stacks. Just because no obvious distortion is observed does not mean that the absolute spacing of the layers has not been affected. Tunable Bragg stacks (as are seen in certain cephalopods) in fact change their reflected color by subtle changes in layer thickness using osmotic changes in water concentration. The authors state that the observed change in reflectance of fixed mirrored guts is likely due the higher index of the fixative, but offer no proof, not even proof that the fixative has a index different enough from seawater to create any observable change. Ideally, the authors should do a sensitivity analysis that looks at how 20% shrinkage affects the models. This should not be hard to do. At the very least, they need to be more candid in the text that shrinkage is a very real possibility that could have affected their results. From personal experience, I know that pelagic animals such as pterotracheids tend to undergo extreme shrinkage upon fixation. The mirrored gut likely doesn't shrink as much, but it's a genuine concern.

Additional comments

no comment

---

## Round 0.3 · accepted · Accept

Thank you for adjusting the manuscript, adding the requested additional analyses and modeling.